# Risk of Pacing-Induced Cardiomyopathy in Patients with High-Degree Atrioventricular Block—Impact of Right Ventricular Lead Position Confirmed by Computed Tomography

**DOI:** 10.3390/jcm11237228

**Published:** 2022-12-05

**Authors:** Patricia Zerlang Fruelund, Anders Sommer, Jens Brøndum Frøkjær, Søren Lundbye-Christensen, Tomas Zaremba, Peter Søgaard, Claus Graff, Søren Vraa, Aksayan Arunanthy Mahalingasivam, Anna Margrethe Thøgersen, Michael Rangel Pedersen, Sam Riahi

**Affiliations:** 1Department of Cardiology, Aalborg University Hospital, 9000 Aalborg, Denmark; 2Department of Clinical Medicine, Aalborg University, 9000 Aalborg, Denmark; 3Department of Radiology, Aalborg University Hospital, 9000 Aalborg, Denmark; 4Unit of Clinical Biostatistics, Aalborg University Hospital, 9000 Aalborg, Denmark; 5Department of Health Science and Technology, Faculty of Medicine, Aalborg University, 9000 Aalborg, Denmark

**Keywords:** computed tomography, pacing-induced cardiomyopathy, heart failure, right ventricular pacing, pacing, atrioventricular block

## Abstract

Prospective studies applying fluoroscopy for assessment of right ventricular (RV) lead position have failed to show clear benefits from RV septal pacing. We investigated the impact of different RV lead positions verified by computed tomography (CT) on the risk of pacing-induced cardiomyopathy (PICM). We retrospectively included 153 patients who underwent routine fluoroscopy-guided pacemaker implantation between March 2012 and May 2020. All patients had normal pre-implant left ventricular ejection fraction (LVEF). Patients attended a follow-up visit including contrast-enhanced cardiac CT and transthoracic echocardiography. Patients were classified as septal or non-septal based on CT analysis. The primary endpoint was PICM (LVEF < 50% with ≥10% decrease after implantation). Based on CT, 48 (31.4%) leads were septal and 105 (68.6%) were non-septal. Over a median follow-up of 3.1 years, 16 patients (33.3%) in the septal group developed PICM compared to 31 (29.5%) in the non-septal group (*p* = 0.6). Overall, 13.1% deteriorated to LVEF ≤ 40%, 5.9% were upgraded to cardiac resynchronization therapy device, and 14.4% developed new-onset atrial fibrillation, with no significant differences between the groups. This study demonstrated a high risk of PICM despite normal pre-implant left ventricular systolic function with no significant difference between CT-verified RV septal or non-septal lead position.

## 1. Introduction

It is well known that right ventricular (RV) pacing may lead to pacing-induced cardiomyopathy (PICM) and heart failure [1]. The detrimental effects have been attributed to the abnormal and dyssynchronous electrical and mechanical activation of the myocardium induced by RV pacing [2]. The myocardial activation pattern is affected by the RV pacing site, and it has been suggested that RV mid- or high-septal pacing may result in a more physiological activation due to the proximity to the specialized conduction system compared to traditional RV apical pacing [3]. However, despite extensive research in the past years, the ideal RV pacing site remains controversial as prospective studies have failed to show a clear benefit from RV septal pacing over non-septal pacing [4,5,6,7]. 

Fluoroscopy is the main imaging modality for guiding pacemaker lead implantation. However, reliable implantation in the intended position is challenged by shortcomings of fluoroscopy. Importantly, RV lead position determined using fluoroscopy has previously been shown inaccurate and poorly reproducible compared with cardiac computed tomography (CT) [8,9]. The limitations of current implantation methods may therefore have resulted in the misclassification of RV lead positions, possibly contributing to the conflicting results from previous studies comparing RV pacing sites. In the real-world pacemaker population, the RV lead is implanted throughout the RV endocardium, providing the opportunity to investigate the effects of chronic pacing from numerous RV lead locations.

The aim of the present study was to investigate the effect of different CT-verified RV lead locations on the risk of PICM in chronically paced patients with advanced atrioventricular (AV) block. 

## 2. Materials and Methods

### 2.1. Study Population

We screened consecutive patients undergoing routine dual-chamber pacemaker implantation due to advanced AV block at Aalborg University Hospital between March 2012 and May 2020 for study eligibility. This was a retrospective study with active clinical follow-up. Patients were eligible for study inclusion if they had a baseline transthoracic echocardiogram (TTE) documenting a left ventricular ejection fraction (LVEF) ≥ 50% prior to pacemaker implantation and frequent RV pacing (≥40%). Exclusion criteria were inability to attend a study-specific follow-up visit (deceased, terminally ill, or no longer followed at the pacemaker outpatient clinic), competing cause of decreased LVEF (severe ischemic heart disease [IHD] or severe valvular heart disease [VHD]), device complications with replacement of the RV lead ≥3 months after implantation, contraindications to iodinated contrast agents for cardiac CT, and unable or unwilling to provide informed written consent to participate. 

### 2.2. Clinical Characteristics and Outcomes

The primary outcome was PICM defined as a ≥10% decrease in LVEF resulting in LVEF < 50% at any timepoint between implant and the end offollow-op. Secondary outcomes were cardiac resynchronization therapy (CRT) device upgrade, new-onset atrial fibrillation, and absolute change in LVEF. Echocardiographic outcomes were obtained by analysis of follow-up TTEs. The remaining characteristics were obtained by review of electronic medical records and pacemaker interrogation reports. Advanced atrioventricular block was defined as second-degree AV block Mobitz type II, 2:1 AV block, higher degree AV block with ≥2 consecutive P-waves not conducted, and third-degree AV block. Severe IHD was defined as acute myocardial infarction confirmed by coronary angiography, requiring revascularization, or having confirmed significant stenosis in ≥2 epicardial arteries requiring revascularization. Severe VHD was evaluated using guideline-recommended quantitative and semi-quantitative measures [10]. 

### 2.3. Pacemaker Implantation and Interrogation 

Pacemaker implantation was performed routinely using biplane fluoroscopy to guide lead implantation. The RV lead was intended for either apical or septal implantation using a manually prepared or preformed stylet. 

Pacemaker interrogation was performed at study follow-up and reports from previous interrogations were retrieved. Information on RV pacing percentage was obtained and reported as the cumulative pacing percentage from implant to study follow-up. 

### 2.4. Cardiac Computed Tomography—Acquisition and Analysis

Available contrast-enhanced thoracic CT scans showing RV lead position were used if already performed for any indication prior to study inclusion. In those with no available CT scan, a contrast-enhanced cardiac CT scan was acquired at the study follow-up visit. A study-specific CT protocol ensuring optimal contrast filling of both ventricular cavities was applied using a second-generation dual source scanner (Siemens Somatom Definition Flash, Siemens Healthcare). The mean total dose length product for the study-specific CT scans was 148.8 ± 102.2 mGy cm and mean contrast dose (Iopromid 370 mg/mL) was 67.3 ± 9.2 mL. The mean slice thickness was 1.1 ± 0.5 mm for the study-specific scans and 1.5 ± 0.5 mm for the clinical scans. 

The CT images were analyzed using a commercially available DICOM viewer providing multiplanar and 3D reconstructions. A simplistic approach was used to analyze RV lead position (Figure 1) [8]. The LV long-axis cavity was divided into equal thirds (apical, mid, and basal) and subsequently the RV lead position was analyzed in short axis view (septal and free wall). Leads positioned at the anterior or inferior septal edges were categorized as septal [11]. For primary outcome analysis, RV lead position was categorized into septal (mid and basal septum) and non-septal (apical septum or free wall). For detailed analysis, septal lead position was further divided into anterior and posterior resulting in four categories: apex, anterior septum, posterior septum, and free wall. RV lead position was independently determined by two experienced outcome-blinded physicians. In case of disagreement, the RV lead position was determined by consensus. 

### 2.5. Echocardiography

All patients had a 2-dimensional TTE performed prior to pacemaker implantation, and a study-specific TTE was repeated at follow-up. Furthermore, all TTEs performed after pacemaker implant and until study follow-up were retrieved. All TTEs were recorded using a 2.5 MHz transducer on a commercially available ultrasound system (VIVID E95, GE Healthcare, Milwaukee, WI, USA) and analyzed using EchoPAC software (GE Healthcare, Milwaukee, WI, USA). LVEF was calculated by two experienced cardiologists blinded to RV lead position using the Simpson’s biplane method and global longitudinal strain (GLS) was based on speckle tracking [12].

### 2.6. Statistical Analysis

Descriptive statistics are reported as mean and standard deviation (SD) or median and interquartile range (IQR) as appropriate for continuous variables. Differences were compared using unpaired Student’s *t*-test or Mann–Whitney test as appropriate. Categorical variables are reported as absolute numbers and percentages and compared using Fisher’s exact test. Modified Poisson regression analysis with robust variance estimation was used to estimate relative risks between groups. For continuous variables, regression was used to estimate differences in means between groups. To overcome potential non-normality and variance inhomogeneity, standard errors were calculated using bootstrap with 5000 replications. No adjusted analyses were performed, as the RV lead was regarded as implanted at random. Hence, confounding through association with both exposure and outcome was considered unlikely. A two-sided *p* < 0.05 was considered statistically significant. STATA version 17 was used to perform all of the statistical analyses. 

## 3. Results

### 3.1. Patient Selection

Of the 1384 patients screened for study eligibility, 608 patients met the inclusion criteria. Subsequently, 181 patients were excluded at baseline due to known severe VHD, severe IHD, contraindications to CT contrast agents, or terminal illness at time of pacemaker implantation, leaving 427 eligible for longitudinal follow-up. Of these, 274 were excluded after baseline due to development of severe VHD, severe IHD, contraindication to CT contrast agents, terminal illness, device complications, unable or unwilling to consent to study participation, or deceased at time of study follow-up. Finally, 153 patients were included for study participation.

### 3.2. CT-Determined RV Lead Position

A study-specific CT scan was performed in 111 patients, as 42 patients had prior thoracic contrast-enhanced CT scans performed for other clinical reasons. Determined from CT, 48 (31.4%) had a septal RV lead position and 105 (68.6%) had a non-septal position. Detailed analysis of RV lead position showed 74 (48.4%) leads located in the apex, 31 (20.4%) leads located on the free wall, 37 (24.2%) leads located on the anterior septum, and 11 (7.2%) leads located on the posterior septum. Only one lead was implanted in the RV basal segment (basal posterior septum), the remaining leads were implanted in the mid or apical segments (Figure 1). 

### 3.3. Clinical Characteristics in Patients with Septal and Non-Septal RV Lead Position

Differences in patient characteristics between the septal and non-septal groups are summarized in Table 1. Baseline characteristics showed no clinically significant differences between the two group. Median duration of follow-up was 3.1 years (range 0.8–8.7 years), and median cumulative pacing percentage was 96.5% (range 45.2–100.0%) with no difference between the two groups. Mean paced QRS duration was significantly longer in the non-septal group compared with the septal group (155 ± 15 vs. 146 ± 17, *p* = 0.003).

### 3.4. Outcomes and RV Lead Position

Of the 153 patients, 47 (30.7%) developed PICM. Nine (5.9%) patients were upgraded to CRT and 22 (14.4%) developed new-onset atrial fibrillation during the study period. There were no statistically significant differences in risk of PICM (RR = 0.89 [95%CI 0.54; 1.46], *p* = 0.63), CRT upgrade (RR = 0.91 [95%CI 0.23; 3.52], *p* = 0.90), or new-onset atrial fibrillation (RR = 0.80 [95%CI 0.36; 1.78], *p* = 0.59) between the non-septal and septal groups (Table 2). 

Overall, a significant 8% reduction in LVEF following pacemaker implantation was observed (95%CI: 7–10%, *p* < 0.01). Among those with PICM, a 19% LVEF reduction (95%CI 17–22%) was observed, and 20 (13.1%) deteriorated to LVEF ≤ 40%. There was no significant difference in LVEF reduction between the septal (10% [95%CI 7–13%]) and non-septal (ΔLVEF = 8% [95%CI 6–9%]) group (*p* = 0.21).

PICM, reduction in LVEF and reduction in negative GLS compared between RV lead implantation in the anterior septum, posterior septum, apex, and free wall are shown in Figure 2. Although the risk of PICM was higher in the posterior septum group (45.5%), there was no significant difference between the groups (*p* = 0.54). Also, the greatest reduction in LVEF was observed in the posterior septum group (15%). However, this again was not statistically significant (*p* = 0.33). Furthermore, no relevant difference in negative GLS reduction (*p* = 0.88) was observed between the groups. 

## 4. Discussion

### 4.1. Main Findings

This is the first study, to our knowledge, investigating the risk of PICM using study-specific contrast-enhanced cardiac CT for the detailed assessment of exact RV lead position. When using cardiac CT as reference, no benefit of septal pacing over non-septal pacing was observed. The study demonstrated an overall high risk of PICM (30.7%) over a median follow-up of 3.1 years despite normal pre-implant LVEF. Clinically relevant, 13.1% had a follow-up LVEF ≤ 40%, 5.9% were upgraded to CRT, and 14.4% developed new-onset atrial fibrillation. 

### 4.2. The Importance of RV Lead Position—Fluoroscopy versus Cardiac CT

Routinely, fluoroscopy has been applied to guide pacemaker lead implantation and determine final RV lead position in studies comparing RV lead positions [13,14]. However, several studies have demonstrated fluoroscopy to be inaccurate when assessing RV lead position [8,9]. Accordingly, using CT, we found a high incidence (20%) of unintended RV free wall implantations in the current study. This is an important finding, as peri-procedural perforation is a risk when implanting in the RV free wall and should therefore be avoided. Misclassification of RV lead positions by fluoroscopy may have contributed to the conflicting results in previous studies. 

Recently, Hattori et al. conducted a retrospective study including pacemaker patients who had an incident thoracic CT performed after pacemaker implantation. RV lead position was classified as septal or free wall and they found an increased risk of cardiac death and heart failure related hospitalizations with RV free wall implantation during 41 months follow-up [11]. However, in the latter study, there were several limitations, e.g., the use of non-specific thoracic CT scans likely with heterogenous scanning protocols for a variety of different clinical indications reducing the quality of the scans for precise assessment of RV lead position. Furthermore, the study evaluated RV free wall pacing versus RV septal pacing, which may be less clinically relevant, as free wall position is not a position one would intentionally aim for when implanting the RV lead. Finally, the exposure group (free wall position) was small counting only 18 (8%) patients and the numbers of events were low. In the current study, we applied a CT scan with ECG-gating and optimal contrast filling of both the right and left ventricular cavities for clear myocardial definition and distinction between RV septum and free wall. We consistently found no advantage of septal lead position over non-septal lead position. Notably, when further dividing RV lead position into anterior and posterior septum, our data showed a trend towards an increased risk of PICM when pacing the RV posterior septum. However, with only 11 patients in this group, this study did not have enough power to conclude on this with certainty and therefore this can merely be considered hypothesis generating. 

### 4.3. Risk of PICM—Comparison with Previous Studies

Previous retrospective studies show a wide variation in reported risks of PICM. This is partly due to discrepancies in PICM definition and may also be related to differences in study designs, duration of follow-up and inclusion criteria [15]. Among patients with normal pre-implant systolic function and pacing percentage ≥20% and using the same PICM definition as applied in this study, Khurshid et al. found a risk of PICM of 19.5 % [16] and Abdin et al. found a risk of PICM of 16.0% [17], which is lower than that reported in our study. Using a more restrictive PICM definition (post-implant LVEF ≤ 40% or CRT-upgrade), Kiehl et al. found a risk of PICM of 12.3% among patients with baseline LVEF > 50% [1] which is comparable to our study where 13.1% had a follow-up LVEF ≤ 40%. The higher risk of PICM in our study may partly be due to inclusion of patients with a higher pacing percentage (≥40%). Furthermore, all patients underwent a prospective study follow-up visit including repeat echocardiogram, overcoming the issue of missing echocardiographic follow-up inherent in many retrospective studies. Other studies have either excluded those without repeat echocardiogram after pacemaker implantation [1,16,17] or have categorized those without repeat echocardiogram as non-PICM [18], likely affecting the reported risk of PICM. 

### 4.4. Limitations

This was a retrospective observational study performed in a single center with a relatively small sample size. Patient selection may have limited the external validity and induced unknown bias, which is why this study can merely be considered hypothesis generating and not conclusive. Importantly, post-treatment selection is not a limitation only for our study. Many of the larger retrospective studies investigating the effect of RV pacing are also limited and likely biased by post-implantation patient selection [1,11,15,16,17,19,20]. The choice of using contrast-enhanced cardiac CT to determine true RV lead position increases patients’ cumulative radiation exposure and can only be applied post-implantation, limiting its use in clinical practice. However, measures were taken to minimize radiation exposure including use of iterative reconstruction algorithms and individual settings of tube current and voltage as well as narrow full current window. Furthermore, patients with a high-quality contrast-enhanced CT scan available with good visualization of the RV lead did not undergo a study-specific scan.

### 4.5. Future Directions

To answer the question of optimal RV lead implantation site, new methods beyond fluoroscopy are needed to ensure intraprocedural accuracy when implanting the RV lead, thus enabling true randomization of RV lead position in prospective studies. It is important to establish a true clinical benefit from RV septal pacing to justify the potential increased procedure duration and the additional fluoroscopy time required for RV septal implantation [4]. 

Also, the simplistic approach used in the current study and previous studies, dividing the RV lead position into arbitrary segments without consideration of individual electroanatomical differences may have limited clinical relevance. Perhaps, the search for an optimal RV lead implantation site needs to move towards physiological pacing including His bundle pacing and left bundle branch pacing [21]. However, widespread use of such methods could have long prospects and may not be relevant for all patients in need of a pacemaker. Additionally, the long-term safety and efficacy of these pacing strategies remain to be evaluated in prospective randomized studies [22]. 

## 5. Conclusions

This study demonstrated an overall high risk of PICM despite normal pre-implant left ventricular systolic function with no significant difference between CT-verified RV septal and non-septal lead position.

## Figures and Tables

**Figure 1 jcm-11-07228-f001:**
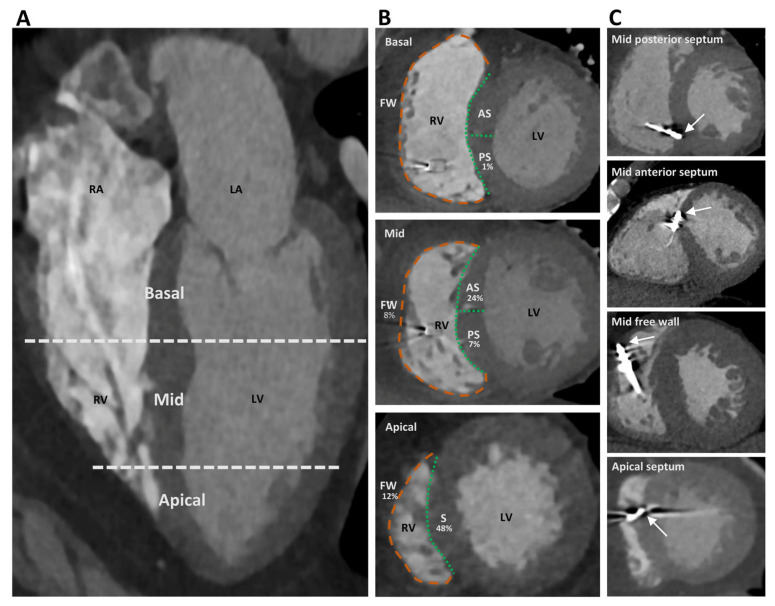
Assessment of right ventricular (RV) lead position. Panel (**A**): A simplistic approach was used dividing the left ventricular (LV) long axis into equal thirds (basal, mid, and apical). Panel (**B**): Septal (S) (green dashed lines) or free wall (FW) (orange dashed lines) positions were then analyzed in short axis view. For primary outcome analysis, the RV lead position was categorized into septal (mid and basal septum) and non-septal (apical septum or free wall). Mid and basal septum were further divided into anterior septum (AS) and posterior septum (PS). The distribution of RV lead positions is shown for each segment. Panel (**C**): Examples of different RV lead tip positions (white arrows). LA, left atrium; RA, right atrium.

**Figure 2 jcm-11-07228-f002:**
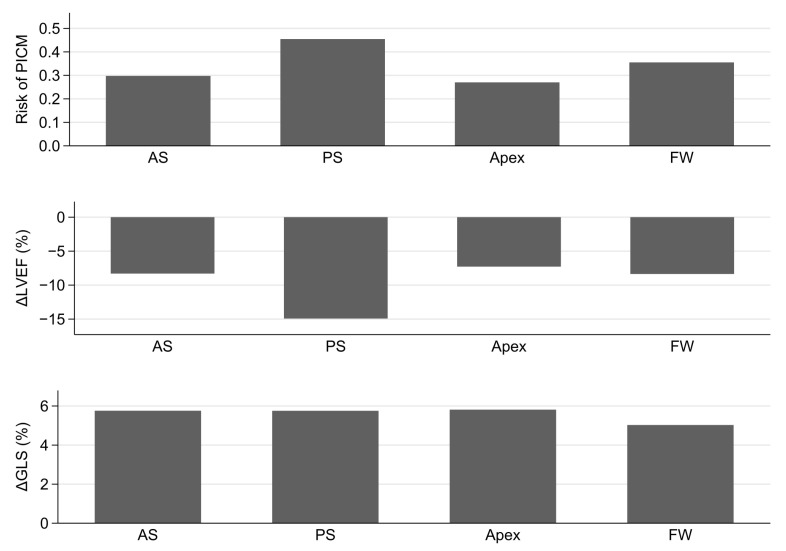
Comparison between anterior septal (AS), posterior septal (PS), apical, and free wall (FW) RV lead position. Upper panel: Risk of pacing-induced cardiomyopathy (PICM). Mid panel: Decrease in left ventricular ejection fract (ΔLVEF) after pacemaker implantation. Lower panel: Decrease in negative global longotudinal strain (ΔGLS) after pacemaker implantation.

**Table 1 jcm-11-07228-t001:** Baseline characteristics in patients with RV non-septal and RV septal lead position.

**Variables**	**Non-Septal (n = 105)**	**Septal (n = 48)**	***p*-value**	**All (n = 153)**
Age (y)	72.9 (64.6–77.7)	72.0 (66.3–75.8)	0.92	72.1 (64.7–76.9)
Male sex	72 (68.6)	31 (64.6)	0.71	103 (67.3)
LVEF (%)	60.2 ± 3.5	60.6 ± 3.8	0.51	60.3 ± 3.6
QRS duration (ms)	119 ± 30	120 ± 29	0.74	119 ± 30
Comorbidity				
Ischemic heart disease	5 (4.8)	7 (14.6)	0.05	12 (7.8)
Valvular heart disease	7 (6.7)	5 (10.4)	0.52	12 (7.8)
Atrial fibrillation	11 (10.5)	4 (8.3)	0.78	15 (9.8)
Hypertension	70 (66.7)	36 (75.0)	0.35	106 (69.3)
Diabetes	24 (22.9)	11 (22.9)	1.00	35 (22.9)
History of smoking	41 (39.1)	24 (50.0)	0.22	65 (42.5)
eGFR (mL/min/1.73 m^2^)	75.3 ± 16.2	73.8 ± 13.6	0.57	74.8 ± 15.4
Medical therapy				
RAS-acting agents	52 (49.5)	30 (62.5)	0.16	82 (53.6)
Betablocker	20 (19.1)	9 (18.8)	1.00	29 (19.0)
Loop diuretics	12 (11.4)	3 (6.3)	0.39	15 (9.8)
Aldosterone antagonist	5 (5.8)	3 (6.3)	0.71	8 (5.2)

Continuous values are reported as mean ± standard deviation or median (interquartile range) as appropriate. Binary values are expressed as numbers (%). eGFR, estimated glomerular filtration rate; LVEF, left ventricular ejection fraction; RAS, renin-angiotensin system; RV, right ventricular.

**Table 2 jcm-11-07228-t002:** Comparison of outcomes in patients with RV non-septal and RV septal lead position.

Outcomes	All (n = 153)	Non-Septal (n = 105)	Septal (n = 48)	RR	95% CI	*p*-value
PICM	47 (30.7)	31 (29.5)	16 (33.3)	0.89	0.54; 1.46	0.63
CRT upgrade	9 (5.9)	6 (5.7)	3 (6.3)	0.91	0.23; 3.52	0.90
New-onset AF	22 (14.4)	14 (13.3)	8 (16.7)	0.80	0.36; 1.78	0.59

Values are given as numbers (%). AF, atrial fibrillation; CI, confidence interval; CRT, cardiac resynchronization therapy; PICM, pacing-induced cardiomyopathy; RR, relative risk; RV, right ventricular.

## Data Availability

Data presented in this study are not publicly available out of consideration for the study participants. Relevant data can be made available upon reasonable request to the corresponding author.

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
