# Peer review of "Risk of Pacing-Induced Cardiomyopathy in Patients with High-Degree Atrioventricular Block—Impact of Right Ventricular Lead Position Confirmed by Computed Tomography"

_jcm, 2022, doi:10.3390/jcm11237228_

Round 1

Reviewer 1 Report

This is a nice retrospective study with active follow up (median 3.1y) on 153 patients with normal baseline LVEF dedicated to unveil the efect of RV lead position on LV sistolic function and PICM apparition.

Minor comments:

- leads on RFW-septal junction may not be all septal as well as not all apical leads are really RFW leads (maybe some information on baseline and paced QRS duration/axis would have been useful to discriminate)

- overall information on baseline and paced QRS duration (and eventually on LV activation time in lead V6) should be available

Reviewer 2 Report

This is a nicely written study, dealing with a still incompletely resolved question: does non-septal RV lead positioning result in more pacing-induced cardiomyopathies than septal pacing. I agree that most previous works comparing apical versus non-apical pacing are flawed by RV lead assessment which was performed by fluoroscopy, which has been proven to be unreliable using classical criteria. Indeed, non-apical pacing in those studies implied “septal” pacing, however we know that it is frequent that leads are in fact positioned on RV free wall when we target the septum. Nevertheless, despite this limitation, most RCTs were either negative or were favoring “septal” pacing, and a meta-analysis of the RCTs favored septal pacing (Shimony et al, europace 2012).

This present study using CT scan with a dedicated protocol for identifying RV lead positioning is welcome.

My questions:

-        You mentioned that you excluded ischemic cardiomyopathies. Did you perform systematical coronary angiogram in patients whose LVEF decreased during follow-up?

-        In your cohort, 42 patients underwent CT-scan for various reasons, thus without dedicated protocol for seeing LV and RV. You mentioned that this was a limitation of Hattori et al (Heart Rhythm 2019) study whose results differed from your study, and I agree that often, it is very difficult to interpret lead positioning in CT scans without proper RV contrast. Why keeping these patients? Was the RV cavity and the septum clearly visible in these CT scans?

-        I don’t understand why the apical third of the septum was considered “non-septal” in your analysis. We can hypothesize that LV activation will be faster with septal-apical pacing than apical-free wall pacing, and LV purkinje can extend quite far towards the apex. As this represent 48% of the implanted leads, this is a very important part of the cohort. Please explain this point. What are the results if you include the septal-apical leads in the “septal” group?

-        An important advantage of septal pacing is the absence of potential for tamponade, compared with free wall or true apical pacing. It should be worth mentioning in the discussion.

Round 2

Reviewer 2 Report

My questions have been answered adequately